# Automated cell counting for Trypan blue-stained cell cultures using machine learning

**Louis Kuijpers**[1,2], **Edo van Veen**[1], **Leo A. van der Pol**[2], **Nynke H. Dekker**[1] *

**1** Delft University of Technology, Delft, The Netherlands, **2** Intravacc B.V., Bilthoven, The Netherlands

* n.h.dekker@tudelft.nl

**Data Availability Statement:** Code and installation instructions: https://gitlab.tudelft.nl/nynke-dekker-lab/public/cell-counter and/or https://gitlab.tudelft.nl/nynke-dekker-lab/public/cell-counter/-/tree/3cb1e0165012402914c5bbdb42179fae0eadcc24

## Abstract

Cell counting is a vital practice in the maintenance and manipulation of cell cultures. It is a crucial aspect of assessing cell viability and determining proliferation rates, which are integral to maintaining the health and functionality of a culture. Additionally, it is critical for establishing the time of infection in bioreactors and monitoring cell culture response to targeted infection over time. However, when cell counting is performed manually, the time involved can become substantial, particularly when multiple cultures need to be handled in parallel. Automated cell counters, which enable significant time reduction, are commercially available but remain relatively expensive. Here, we present a machine learning (ML) model based on YOLOv4 that is able to perform cell counts with a high accuracy (>95%) for Trypan blue-stained insect cells. Images of two distinctly different cell lines, *Trichoplusia ni* (High Five™; Hi5 cells) and *Spodoptera frugiperda* (Sf9), were used for training, validation, and testing of the model. The ML model yielded $F1$ scores of 0.97 and 0.96 for alive and dead cells, respectively, which represents a substantially improved performance over that of other cell counters. Furthermore, the ML model is versatile, as an $F1$ score of 0.96 was also obtained on images of Trypan blue-stained human embryonic kidney (HEK) cells that the model had not been trained on. Our implementation of the ML model comes with a straightforward user interface and can image in batches, which makes it highly suitable for the evaluation of multiple parallel cultures (e.g. in Design of Experiments). Overall, this approach for accurate classification of cells provides a fast, bias-free alternative to manual counting.

## Introduction

In an era characterized by a growing interest in proteins, drugs, antibodies, and enzymes for therapeutic and biochemical applications, the significance of their production platforms has increased substantially. These platforms encompass a diverse range of cell lines, spanning from bacterial to human cells, each with their own advantages and disadvantages, and have assumed paramount importance in modern biotechnology [1–5]. All these cell-based production platforms share a common requirement for precise, accurate and rapid cytometric measurements. In this study, we conducted an investigation on two widely used cell types: insect cells and HEK cells, which serve as foundational components of two highly prevalent expression systems, namely the baculovirus expression vector system (BEVS) [5–7] and a

(persistent URL) Annotated test, train, and validation data sets, the complete set of evaluated images (n=122 for insect cells and n = 52 for HEK cells) and confusion matrices for determination of the performance parameters: doi.org/10.4121/21695819.

**Funding:** Recipients: L.v.d.P. and N.H.D. Grant name: COMET Grant number: none Funder: Intravacc B.V. Funder website: https://www.intravacc.nl/ The funders had no role in study design, data collection and analysis, decision to publish, or preparation of the manuscript.

**Competing interests:** The authors have declared that no competing interests exist.

mammalian expression platform [8, 9]. To maximize the protein expression using the BEVS or HEK cells, maintaining the cell culture in a viable state is critical [10].

Conventional determination of cytometric parameters, most prominently, alive and intact dead cell concentrations is performed using a hemocytometer (e.g. Bürker-Türk counting chamber) [11]. The chamber consists of nine squared wells, each with a set volume, and is placed under a brightfield microscope (**Fig 1**). Dividing the total cell count by the total volume of the counted large squares yields the cell density. To enhance the accuracy of counting, dye exclusion methods are used to assess whether a cell is alive or dead: because dead cells lose membrane integrity, dye can traverse the membrane, staining dead cells in a distinctive color [12]. The most commonly used dye for insect cells is Trypan blue [13]. The major advantage of this conventional approach is the direct observation of the cell culture by the operator, which enables the rapid detection of problems such as contamination or aggregation of cells. However, manual cell counting is time-consuming, especially when a multiple counts are performed on the same culture for increased accuracy. Additionally, because this technique relies on human interpretation, operator-to-operator variance is not uncommon. Lastly, the reproducibility of manual cell counts can be low, especially if high cell density cultures are used [14].

Some of the human errors inherent in conventional cell counting approaches can be avoided using automated cell counters (**Fig 1**). Many research groups have transitioned to this more reproducible and less time-consuming technique [15–20]. Unfortunately, at present this technology comes with its own limitations. For example, the majority of the slides used for automated cell counts contain open experimental handling, meaning that they cannot be applied to infected cultures. Moreover, new automated cell counters range in price from €5,000 to €25,000, creating considerable investment cost for R&D labs [21]. This holds especially true for research groups keen to explore the BEVS as a possibility for specific protein expression but not yet committed to the technology in the longer term. Lastly, the automated cell counters based on BF-microscopy can have a significant error range, particularly, when cells are aggregated or cultures reach high cell densities.

Here, we have developed a machine learning (ML) model based on the publicly available YOLOv4 model, together with a user-friendly interface, for the determination of dead and alive insect cells [22]. Our aim was to maximize the accuracy with an error margin of at most 5%. ML models have significant advantages over other automated cell counting software packages (e.g. ImageJ). Firstly, a good ML model makes manual image (pre-)processing obsolete [15, 17]. Secondly, enabling batch image processing could reduce significantly overall process time in the evaluation of multiple cultures (e.g. during Design of Experiments) (**Fig 1**). Such advantages have been deployed in the counting of red blood cells and several other cell lines; however, to date an appropriate model for the counting of insect cells remains lacking [19, 23–25].

## Methods

### Cell culturing & preparation for manual cell counting

The two most commonly employed insect cells were used to train and test the ML model: *Trichoplusia ni* (Gibco High Five™; Hi5 cells) and *Spodoptera frugiperda* (Gibco; Sf9). The Hi5 and Sf9 cells were separately cultured in final volumes of 50 ml in 125 ml Erlenmeyer culture flasks using Sf900™ II Serum-free medium (Gibco), and seeded with a cell density of $7.5 * 10^4$ cells/ml and $5.0 * 10^5$ cells/ml, respectively. Samples (~200 µl) were taken every 24 h for 4 d. The cell culture samples were mixed 1:1 with Trypan blue (Gibco), and 10 µl of each resulting mixture was transferred to a Countess cell counting chamber slide (Invitrogen). The counting chamber slides were placed under a brightfield microscope, and ~10 fields of view were imaged per counting chamber at a 10X magnification using an Olympus CKX41 microscope

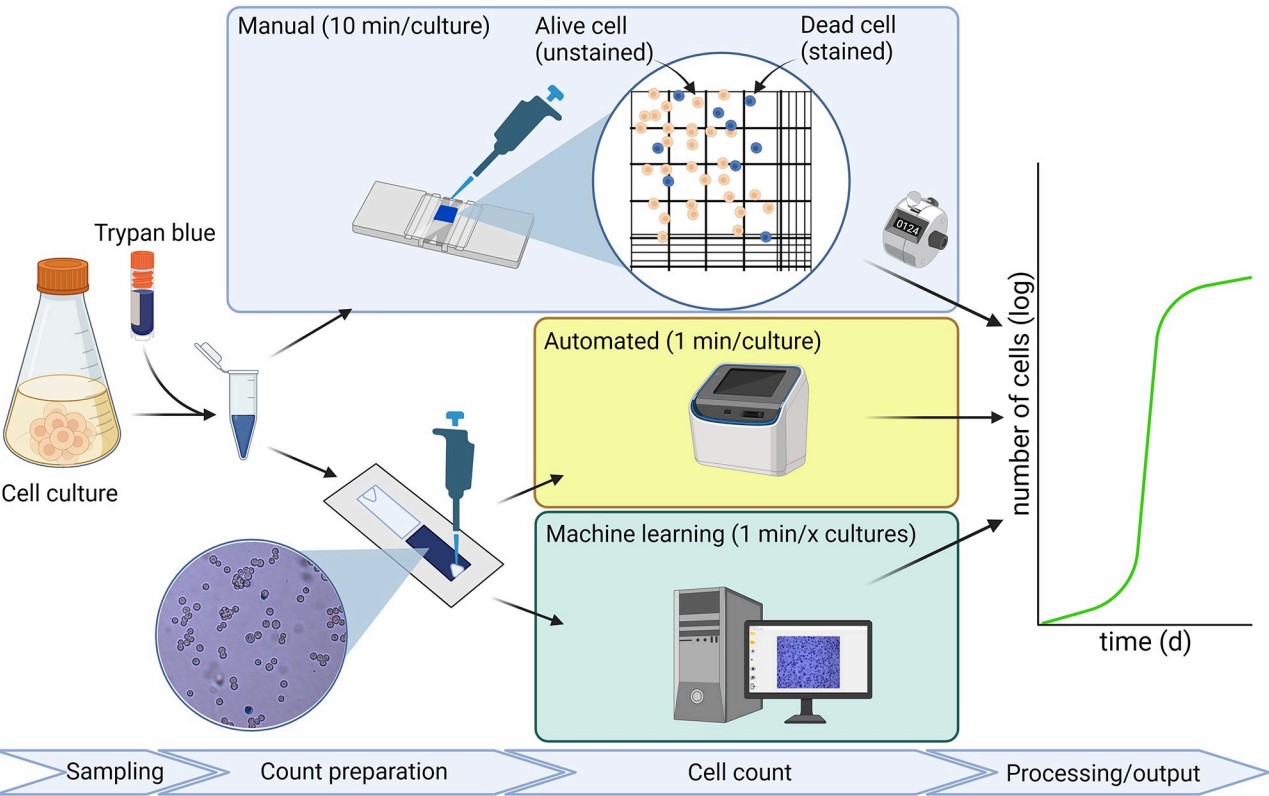

**Fig 1. Overview of the most commonly used techniques for cell counting.** In cell counting, one exploits the difference in membrane integrity between alive (membrane intact, no staining of the cell interior; thus cell observed as white) and dead cells (membrane damaged, allowing Trypan blue to traverse the membrane; thus cell observed as blue). Tracking alive cell density over time then allows one to establish the growth curve of the cell culture. Conventional cell counting (depicted in light blue box) using a counting chamber represents a time-consuming protocol susceptible to significant operator-to-operator variance. Automated cell counters (depicted in yellow box) significantly reduce the operation time, at the expense of a loss of direct observation of the cell culture by the operator and a high investment cost. Our ML model for cell counting (depicted in green box) provides the operator with a cell counting approach of low investment cost that is rapid and accurate, and permits both the direct observation of cell culture and the opportunity to perform cell counts on infected cultures. Created with BioRender.com.

(providing a 0.96 mm$^2$ field of view per image) and cellSens image acquisition software package (Olympus).

Human embryonic kidney cells (HEK cells) were cultured in 1 l bioreactors using BalanCD HEK293 medium (Irvine Scientific). Images were captured after 2 and 3 d of growth. Sampling, sample preparation, and imaging was performed analogously to that of the insect cells.

## Baseline ML model

The cell counter model was based on the You Only Look Once version 4 (YOLOv4) object detection model [22]. The YOLOv4 model consists of three architectural components referred to as the head, neck, and backbone. The backbone is the initial part of the neural network responsible for extracting the feature maps from the input image. In YOLOv4, the backbone is based on the CSPDarknet53 architecture and employs a Cross Stage Partial network (CSPNet) structure to improve information flow and feature extra efficiency in comparison to its predecessor: Darknet53 [22, 26]. CSPDarknet53 contains a series of convolutional layers that gradually downsample the input image, capturing features at difference scales and levels of abstraction. The extracted features are then passed to the subsequent stages of the model. The neck is an intermediary component inserted between the backbone and the head of the

network. It plays a crucial role in aggregating features from multiple stages of the backbone. In YOLOv4, the neck architecture is designed to enhance feature fusion and information flow. It consists of bottom-up pathways, which capture high-resolution features from different stages of the backbone, and top-down pathways which upsample and fuse these features to create a rich feature representation that is used by the detection head [22, 27, 28]. The head is the final part of the network responsible for predicting object classes and bounding boxes. It takes the fused and aggregated feature maps from the neck and applies additionally convolutional and fully connected layers to predict the presence of objects, their classes, and the corresponding bounding boxes. The head architecture is designed to accurately locate and classify objects in the image [22, 29]. Distinguishing itself from alternative object detection methods, YOLO's distinctive trait lies in its single-pass approach to image analysis, which lends it greater computational efficiency. The YOLOv4 model was pre-trained on the MS COCO data set [30, 31].

## Training the model

Data sets were annotated using labelImg, a free, open source tool for graphically labelling images [32]. A training set (46 insect cell images containing 5022 alive cells and 938 dead cells) and a validation set (14 insect cell images containing 2345 alive cells and 510 dead cells) were constructed, and the cells were annotated manually as dead or alive using the aforementioned tool. The training set images contained cell densities ranging from 5 to > 300 cells per field of view (FoV) of 0.96 mm$^2$. For the training of the original YOLOv4 model in order to obtain our optimized, insect cell dectection ML model, we used an Adam optimizer with an initial learning rate of $2 * 10^{-4}$ and a final learning rate of $1 * 10^{-6}$. The number of iterations through the data set (epochs) was set to 100. Our ML model was chosen by picking the model for which the validation set loss was the lowest ("early stopping"), to prevent overfitting. The data sets used [33] and further details of the implementation [34, 35] are provided separately (S1 File).

## Post-processing

The model used three post-processing parameters to produce a final result, namely the confidence score threshold (default = 0.4); the intersection over union threshold (default = 0.3); and the bounding box (bbox) size threshold (default = 200 px$^2$). The default values were the values used during training/validation. The values of the three post-processing parameters could be adapted to make the model more or less sensitive, with the following consequences. The model will return as a prediction any object with confidence score larger than the confidence score threshold. Decreasing this threshold will lower the threshold for positive object classification, and hence in our context will return more cells at the expense of an increased risk of false positives. The intersection over union threshold dictates how much overlap is allowed between neighboring objects. Lowering this threshold will allow for bounding boxes to be closer together, possibly leading to more detected cells, especially aggregated or slightly deformed ones. However, it also risks false double detections of a single cell. Finally, the bbox size threshold parameter filters out any detections with a small bbox size. Modifying this parameter could be beneficial upon changes in the magnification or cell size; increasing the value of this post-processing parameter will increase the size of the allowed bounding boxes, and vice versa.

## Model performance

The model performance was evaluated by the construction of a confusion matrix from a test data set consisting of 122 images, containing >20,000 alive and dead cells. Images of the test data set were newly acquired and not part of the train and validation data sets. The test set was used in order to understand how well the model functions for different cell strains, cell

densities, and viabilities. The numbers of true positives (TP), false positives (FP), and false negatives (FN) on the test set were determined manually [36, 37]. Three metrics were computed to evaluate the model performance. The recall (alternatively referred to as sensitivity or true positive value, *TPV*) measures the extent of error caused by false negatives, and is defined as:

$$TPV = \frac{TP}{TP + FN}. \tag{1}$$

The precision (or positive predictive value, *PPV*) measures the extent of error caused by false positives, and is defined as:

$$PPV = \frac{TP}{TP + FP}. \tag{2}$$

The *F*1 score, the leading performance indicator for ML models, is defined the harmonic mean of the precision and the sensitivity:

$$F1 = \left(\frac{PPV^{-1} + TPV^{-1}}{2}\right)^{-1} = \frac{2TP}{2TP + FP + FN}. \tag{3}$$

## Results & discussion

To verify that the ML model was accurate for a broad range of Trypan blue-stained cell images, we tested the model on a large data set containing different cell strains, cell densities, and viabilities. The model was tested using a data set of 122 images (66 images from Hi5 cell cultures and 56 images from Sf9 cell cultures) containing 20,046 cells. The majority of the images were captured successively and in a single experimental procedure at 10X magnification, then batch processed by the ML model. The model classified cells as either alive or dead and assigned them blue or red bounding boxes, respectively (**Fig 2**). The post-processing thresholds for the confidence interval, intersection over union, and bounding box size (see section Post-processing) were set to 0.3, 0.3, and 25 px$^2$, respectively. These post-processing thresholds, which were optimized by visual inspection, were dependent on cell fitness and density, as well as on microscope settings such as the magnification. Based on visual inspection we could determine that

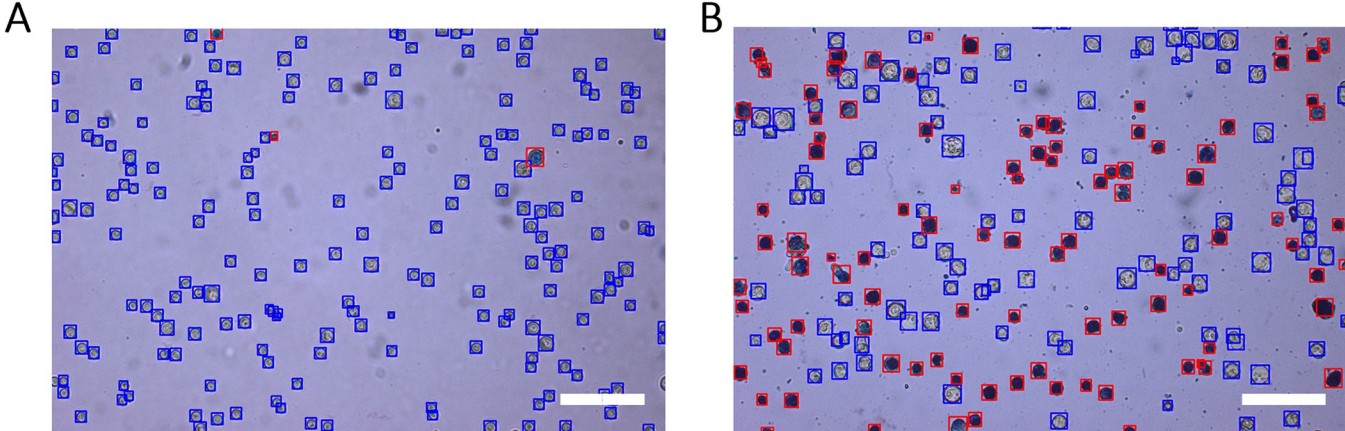

**Fig 2. ML model classified cells (blue = alive, red = dead) in images originating from test data set.** Images of Hi5 cell cultures taken at 10X magnification with their contents classified by the ML model. Alive cells were indicated by blue bounding boxes and dead cells by red bounding boxes. Scale bar represents 250 µm. (A) Hi5 cell culture grown for 4 d at 28˚C in Sf900iiSFM. (B) Hi5 cell culture 7 days post infection with baculovirus at an MOI of 0.01.

the model was able to identify the presence of the majority of the cells and classify them correctly. Furthermore, the model did not classify as cells any of the cell debris and/or protein aggregation that was visible as small blue dots (**Fig 2B**), thus obviating additional manual image processing. Additionally, anomalies originating from microscope contamination, such as scratches or dust, were not classified as cells. Relative to previously reported automated cell counting models, which depended on conventional image processing to eliminate such contaminants, the ML model was fast and reduced the overall process time for cell count determination [15–17, 38].

A first qualitative assessment of the ML model's performance in classification was assessed by comparing it to multiple independent manual inspections of all 122 classified images. The result is summarized in correlation plots (**Fig 3A and 3B**) in which the black dotted diagonals represent the ideal scenario in which the ML-determined and manually determined cell counts are identical. Over the entire range from 5 to >400 cells per FoV, the machine learning model was able to determine the alive cell counts with very high accuracy ($r^2 > 0.99$ from linear fits (red lines) to the data) for both Hi5 (**Fig 3A**) and Sf9 cells (**Fig 3B**). These correlation coefficients, obtained from the linear fit through the data, are similar or higher than previously obtained using image processing software packages or ImageJ [15–17]. High accuracies were also achieved by the ML model for dead cell counts for Hi5 (**Fig 3C**) and Sf9 (**Fig 3D**) cells, with a maximum overestimation of ~4.5% for the higher cell counts (>150 cells/FoV), originating from the double counts of aggregated cells. Aggregated cells can result in a significant overestimation of cell counts due to the lack of cell boundary detection, leading to double counts. This is in line with the decreased accuracy observed for heavily aggregated cultures by manual cell counting methods and the majority of automated cell counters [39]. In general, dead cell detection and classification represents a greater challenge. This is because these cells have less distinct features (e.g. a less pronounced outer boundary and decreased distinction from the blue background relative to white alive cells), and are more susceptible to disintegration as a result of apoptosis. Indeed, the accuracy of (intact) dead cell counts achieved by the ML model exceeded that of previously published work [15–17]. Because the majority of the test images were acquired from cultures with conditions supporting cell growth, the dead cell counts were low relative to the alive cell counts. It needs to be noted that the accuracy of all cell counting techniques based on cell phenotype increases when the sample is derived from cultures exhibiting low levels of cell lysis. These techniques only take into account alive and intact dead cells, and therefore cannot determine the degree of cell lysis. Any extent of cell lysis will necessitate further evaluation, such a through the measurement of lactate dehydrogenase levels [40]. To assess whether the ML model could accurately determine the number of dead cells even at low cell counts, we magnified this regime (insets **Fig 3C and 3D**). For both cell lines the ML model was able to determine the cell counts with high accuracy. The viability of the cell cultures was determined based on the viable and total cell counts in the individual images and categorized per cell type (**Fig 3E and 3F**). Again, as most of the cell images originated from cultures with conditions supporting cell growth, the higher viabilities are overrepresented in the data set (insets **Fig 3E and 3F**). For both cell lines jointly, on average the difference in viability determined by manual counting and the ML model was 0.99% ± 1.55%, which is lower than the average operator-to-operator variation in manual viability determination using a counting chamber (2–13%) [41] and well within the target (maximum difference of 5%) of this study.

As a second, more quantitative assessment of the performance of our ML model, we constructed a confusion matrix to determine the *F*1 score, sensitivity, and recall of all images, both per cell line and for both cell lines jointly (**Fig 4**) [36, 37]. The per cell line quantification was performed to reveal any differences in the performance of the model between them. For the

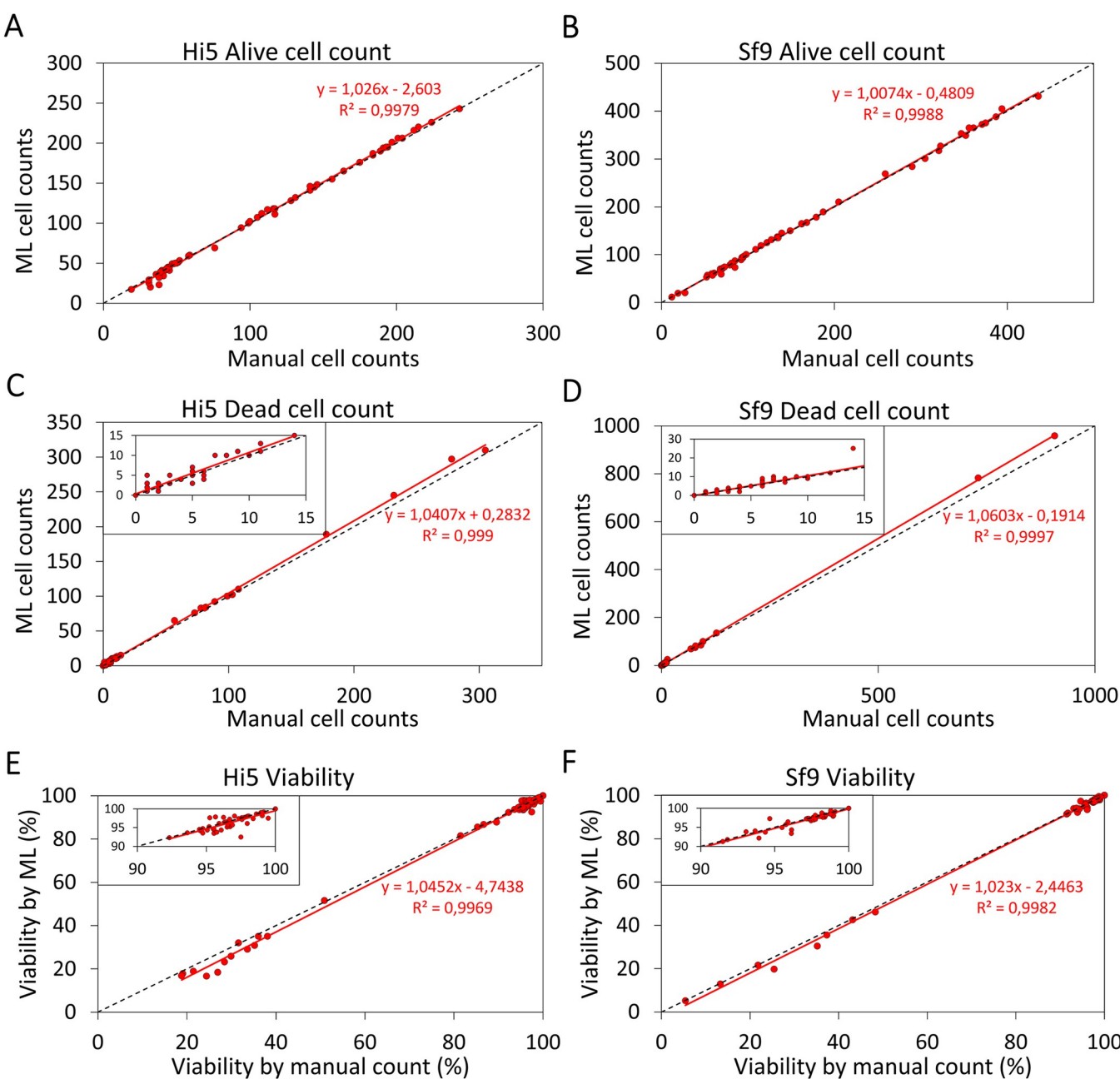

**Fig 3. Correlation plots between the manual cell count of images and the cell count by the ML model.** (A) Alive cell counts for Hi5 cells (n = 66). (B) Alive cell counts for Sf9 cells (n = 56). (C) Dead cell counts for Hi5 cells (n = 66). Inset: magnification of the region with lower dead cell counts. (D) Dead cell counts for Sf9 cells (n = 56). Inset: magnification of the region with lower dead cells counts. (E) Calculated viability of Hi5 cell cultures. Inset: magnification of the region with higher viabilities. (F) Calculated viability of Sf9 cell cultures. Inset: magnification of the region with higher viabilities.

alive cell count, this analysis showed that the ML model performed better for Hi5 ($F1$ = 0.99) cells than for Sf9 ($F1$ = 0.96) cells. This increased performance most likely originated from the larger size of the Hi5 cells, which made them easier to detect [42]. For both cell lines jointly, the model was able to detect alive cells (**Fig 4A**) slightly better than dead cells (**Fig 4B**), a consequence of their greater consistency in size and shape and the larger number of alive cells in the training data set. For all investigated conditions (per cell line and jointly; alive and dead cells),

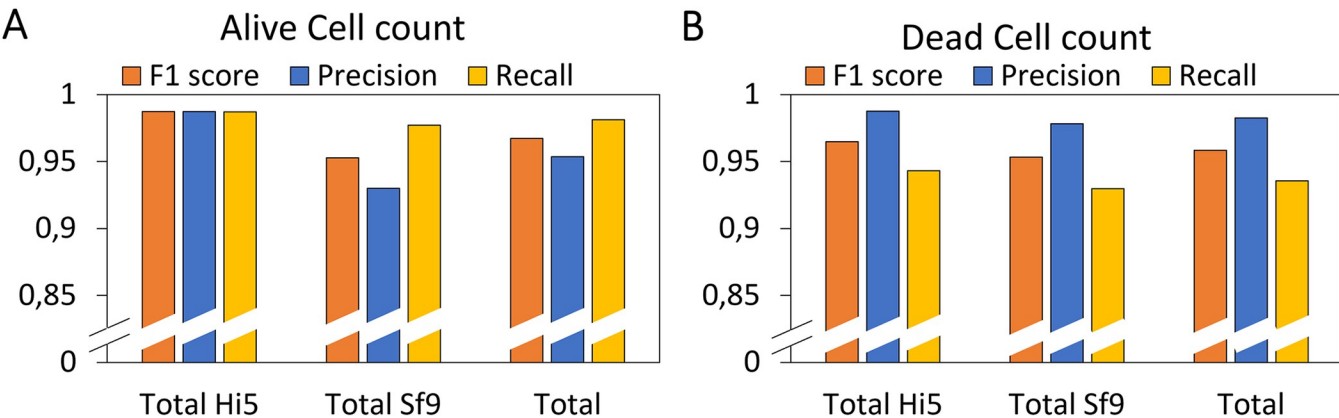

**Fig 4. Performance of the ML model based on confusion matrix theorem.** The independent performance parameters were calculated using Eqs (1)–(3) for Hi5 (n = 66 images) and Sf9 (n = 56 images) data sets both separately, and jointly (totaling 20,000+ cells in 122 images). False positives, false negatives, and true positives were determined manually.(A) Performance parameters for the alive cell counts both per cell line and jointly. (B) Performance parameters for the dead cell counts both per cell line and jointly.

the leading performance parameter ($F1$ score) exceeded 0.95. These high $F1$ scores demonstrate that this ML model is highly suitable for accurate cell density and viability determination of cell cultures.

To examine whether there were differences in performance between images with a high and low cell count, we investigated the $F1$ scores across the range of cell densities (**Fig 5**). As we can observe in both panels, we note that the ML model is able to accurately classify both alive and dead cells in images with a very high cell density ($F1 > 0.90$ for cell counts >400 cells/FoV). This allows one to apply the ML model to cell cultures with a high cell density without the need for additional dilution prior to analysis, thus reducing additional experimental error and process time, provided that aggregation is not apparent. If it is, then dilution is required, analogously to other cell detection methods (e.g. manual and automated cell counting). The ML model yields lower $F1$ scores for both alive (**Fig 5A**) and dead (**Fig 5B**) cell counts at lower cell counts, with this effect being more substantial for the dead cell counts (inset to **Fig 5B**). A decrease of the $F1$ scores with lower cell counts is inherent to their

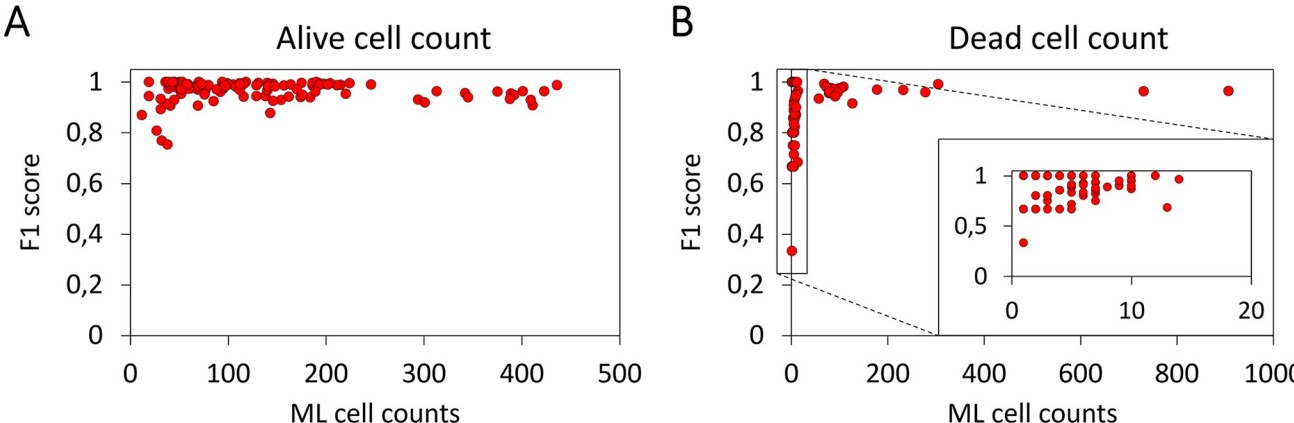

**Fig 5. Investigation of a drop-off in F1 scores at different insect cell counts per image.** Sf9 and Hi5 data sets combined. (A) *F1* scores for alive cell counts per FoV assigned by the ML model (n = 122). (B) *F1* scores for dead cell counts per FoV assigned by the ML model (n = 122). Inset: magnification of the region with low cell counts.

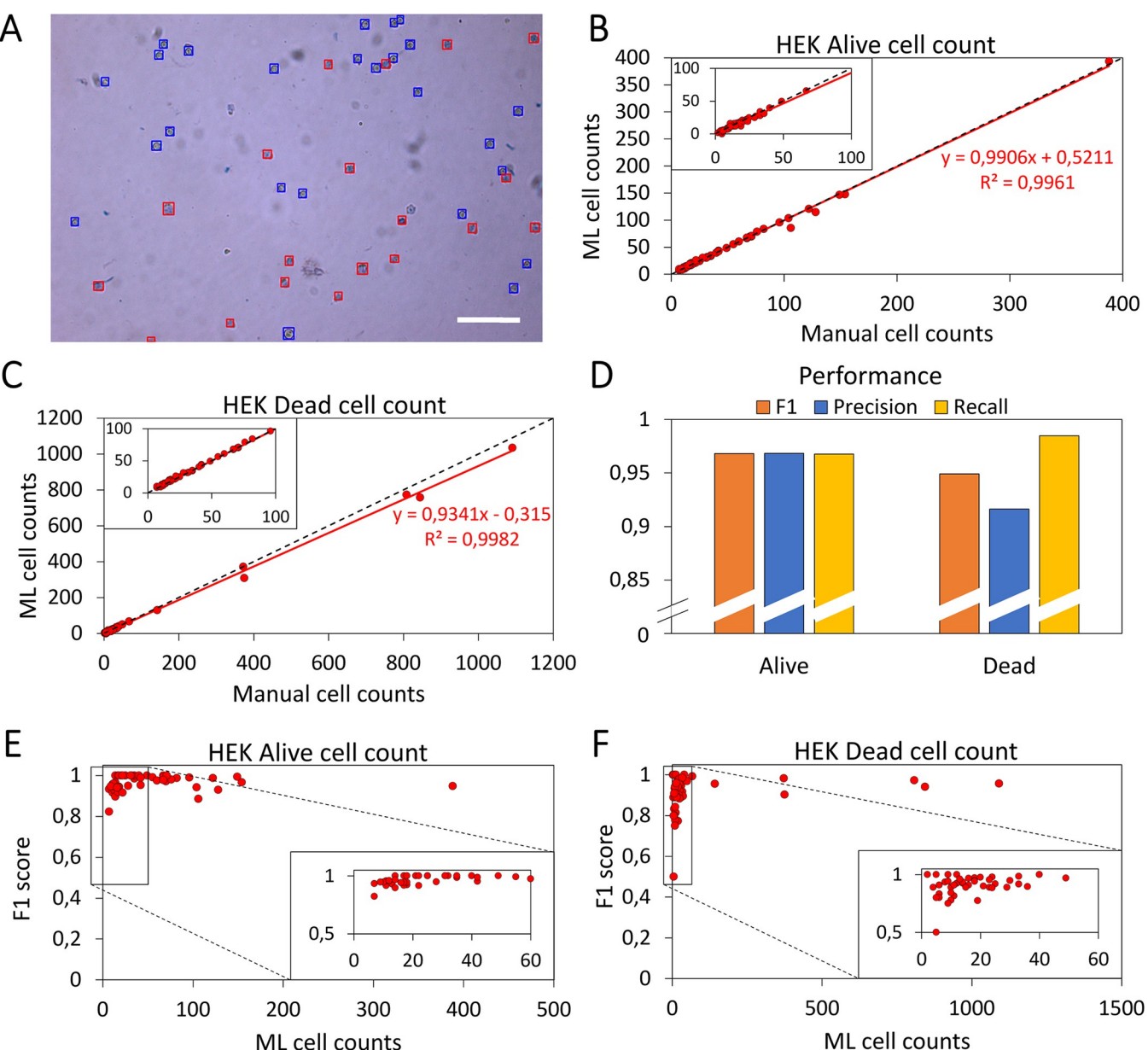

**Fig 6. Evaluation of the performance of the ML model on HEK cells.** (A) Image of HEK cell culture taken at 10X magnification with contents classified by the ML model. Alive cells are indicated by blue bounding boxes and dead cells by red bounding boxes. Scale bar represents 250 μm. (B) Correlation plot between the alive cell counts determined manually and by the ML model on HEK cells. Inset: magnification of the lower count regimes. (C) Correlation plot between the dead cell counts determined manually and by the ML model on HEK cells. Inset: magnification of the lower count regimes. (D) *F*1 scores of the alive cell counts assigned by the ML model for HEK cells (n = 52). Inset: magnification of the low cell count regime. (E) *F*1 scores for dead cell counts assigned by the ML model for HEK cells (n = 52). Inset: magnification of the low cell count regime. (F) False positives, false negatives, and true positives were determined manually. The independent performance parameters were calculated over all 6801 HEK cells in all 52 images.

computation: for a low number of cells (TP in Eq (3)), the effect of a single misclassification (either FP or FN in Eq (3)) has a disproportionate influence. This, together with the low total number of dead cells per FoV detailed above, accounts for the more severe decrease in *F*1 scores observed at lower cell counts for dead cells.

Given the promising performance of our ML model for insect cells, we next investigated the applicability of the same model to Trypan blue-stained human embryonic kidney (HEK)

cells. These cells exhibit a distinctive differentiation pattern similar to that of insect cells, where white and blue markers are used to distinguish between alive and dead cells, respectively. As before, images were taken of cultures with conditions supporting cell growth, and these were then analyzed using slightly altered post-processing parameters (confidence interval, intersection over union, and bounding box size were set at 0.15, 0.1, and 25 $px^2$, respectively) set following visual inspection of the initial classification by the model using default settings. The decrease in the values of the confidence interval and intersection over union parameters followed logically from the smaller size of HEK cells relative to insect cells (**Fig 6A**).

The correlation plots for HEK cells (**Fig 6B and 6C**) indicated similar performances of the ML model for alive (**Fig 6B**) and dead (**Fig 6C**) cell counts, with a marginal underestimation for the higher dead cell counts (>200 cells/FoV). Anew, the performance parameters determined that the ML model was able to determine the majority of the cell counts correctly (**Fig 6D**; $F1 = 0.97$ and $0.95$ for alive and dead cell counts, respectively), however, marginally lower than for insect cells. This decreased performance originated from the much smaller size (diameter: 11–15 μm) of HEK cells compared to insect cells (diameter: 17–30 μm), their less pronounced cell boundaries, and most prominently, the lack of training of the ML model on these cells [8, 42]. Similarly to insect cells and especially so for dead cell counts, we observed drop-offs in the $F1$ score for lower cell counts (**Fig 6E** and **6F**). The rationale behind this drop-off is analogous to the abovementioned detrimental effects of misclassification at low cell counts on the $F1$ score.

## Conclusions

Here, we have presented an alternative to manual or automated cell counting, for Trypan blue-stained microscopic images of insect cells and HEK cells. Our ML model was able to determine cell counts with very high accuracy, without the need for manual image processing prior to automated cell counting. Our ML model provides a faster and cheaper alternative to manual cell counting and acquisition of a commercially available automated cell counter, respectively. Moreover, the model is able to quickly process a large number of images, allowing the operator the opportunity to rapidly determine the cell counts of multiple cultures. We expect that this ML model will provide a useful tool for operators of cell cultures at all scales, and that it will lower the threshold for scientists requiring protein expression by insect cell production platforms.

## Supporting information

**S1 File. Supplementary information.**
(DOCX)

## Acknowledgments

We thank Bèr van der Wijden for providing the HEK cells for imaging, Abbas Freydoonian for maintaining the insect cell cultures, and Oussama Akhiyat for aiding in the code development.

## Author Contributions

**Conceptualization:** Louis Kuijpers, Edo van Veen.

**Data curation:** Louis Kuijpers.

**Formal analysis:** Louis Kuijpers.

**Funding acquisition:** Nynke H. Dekker.

**Investigation:** Louis Kuijpers, Edo van Veen.

**Methodology:** Louis Kuijpers, Edo van Veen.

**Project administration:** Leo A. van der Pol, Nynke H. Dekker.

**Resources:** Leo A. van der Pol, Nynke H. Dekker.

**Software:** Louis Kuijpers, Edo van Veen.

**Supervision:** Edo van Veen, Leo A. van der Pol, Nynke H. Dekker.

**Validation:** Louis Kuijpers, Edo van Veen.

**Visualization:** Louis Kuijpers, Nynke H. Dekker.

**Writing – original draft:** Louis Kuijpers, Edo van Veen.

**Writing – review & editing:** Leo A. van der Pol, Nynke H. Dekker.

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
