## [Decision Letter · Decision Letter 0]

18 Jul 2023

PONE-D-23-16673Automated cell counting for Trypan blue-stained cell cultures using machine learningPLOS ONE

Dear Dr. Dekker,

Thank you for submitting your manuscript to PLOS ONE. After careful consideration, we feel that it has merit but does not fully meet PLOS ONE’s publication criteria as it currently stands. Therefore, we invite you to submit a revised version of the manuscript that addresses the points raised during the review process.

We look forward to receiving your revised manuscript.

Kind regards,

Abdul Rauf Shakoori

Academic Editor

PLOS ONE

“We thank Bèr van der Wijden for providing the HEK cells for imaging, Abbas Freydoonian for maintaining the insect cell cultures, and Oussama Akhiyat for aiding in the code development. We gratefully acknowledge funding from Intravacc B.V. to L.v.d.P. and N.H.D.”

“Recipients: L.v.d.P. and N.H.D.

Grant name: COMET

Grant number: none

Funder: Intravacc B.V.

Funder website: https://www.intravacc.nl/

5. We note that Figure 1 in your submission contain copyrighted images. All PLOS content is published under the Creative Commons Attribution License (CC BY 4.0), which means that the manuscript, images, and Supporting Information files will be freely available online, and any third party is permitted to access, download, copy, distribute, and use these materials in any way, even commercially, with proper attribution. For more information, see our copyright guidelines: http://journals.plos.org/plosone/s/licenses-and-copyright.

Reviewers' comments:

Reviewer's Responses to Questions

**Comments to the Author**

1. Is the manuscript technically sound, and do the data support the conclusions?

Reviewer #1: Partly

Reviewer #2: Yes

Reviewer #3: Partly

2. Has the statistical analysis been performed appropriately and rigorously? 

Reviewer #1: Yes

Reviewer #2: Yes

Reviewer #3: Yes

3. Have the authors made all data underlying the findings in their manuscript fully available?

Reviewer #1: No

Reviewer #2: Yes

Reviewer #3: No

4. Is the manuscript presented in an intelligible fashion and written in standard English?

Reviewer #1: Yes

Reviewer #2: Yes

Reviewer #3: Yes

5. Review Comments to the Author

Reviewer #1: Review: Automated cell counting for Trypan blue-stained cell cultures using machine learning

The authors present ML-based application for automated cell counting that provides high accuracy at low cost. This research is significant since manual count using traditional approaches such as hemocytometer is often unreliable and not reproducible as results may vary when analyzed by different operators. The manual binary classification live or dead is often hard to make since sometimes cells are not just “white” and “blue” but in between and some operators may call it live and some dead.

I appreciate the efforts done by the authors of this manuscript since they are aimed to solve a problem for counting insect cells that lack quantitative and reliable tools. They also aim to avoid buying expensive technology, which is highly valuable. Overall, the work presented is technically sound and promising. I only have concerns that they seemed to verify their model using manual human visual approach which they criticized themselves. I think their manuscript deserves publication, but needs to address at least the following: 1) how many operators verified the model manually and if there was 100% consensus or not, 2) It seems that there are out-of-focus cells in the pictures. Could a different, thinner chamber be used to make sure all the cells are in the same focus plane?, 3) It would be best if they tested or trained control samples, dead cell control sample, for example, is easy to prepare, since they can use heat-shock or fixation protocols to produce stable dead cells. I think this is far more important than mixing cultures, though, I appreciate the value of the mixed cell population as well. 4) It seems that model can be improved by training close to equal # of live and dead cells to avoid issues discussed in the manuscript (model was able to detect alive cells better than dead cells). Why was it not done if they recognized the issue? 5) I was not clear if their approach solved the aggregation problem. It seems that it did not, but they emphasized that it is a problem and it was my understanding that they are also trying to solve that issue with their ML technique.

Suggestions/major concerns:

1) The title of the manuscript is about cell counting and trypan blue assay. Thus, the introduction should start by introducing state-of-the art cell counting and any issues with the measurements that the authors are aiming to solve. I was lost how for a bit reading about protein expression etc since they didn’t link it to cell counting until the fourth paragraph.

2) Figure 1, Machine learning approach, it is unclear what is the sample. Did the authors use a hemocytometer and brightfield images for the analysis or the chambers for automated cell counters. Later the information can be found in the methods, but Fig.1 comes earlier than the explanation. I suggest adding at least a cartoon image of the slide+microscope to the ML method part like for the others.

3) For the training of the model, wouldn’t it be beneficial to prepare control samples of healthy cultures (with only few dead cells) and 100% dead cell control sample by treating them with a known cell killing methods (e.g. heat-shock or fixation)? Also, for the training of the model, there is disproportion between the live cell # and dead cell #. The ML experts typically tell me that we need to train equal number of instances in each class to avoid imbalances/bias in the training. This limitation is pointed out later on line 260 “For both cell lines jointly, the model was able to detect alive cells (Figure 4A) slightly better than dead cells (Figure 4B), a consequence of their greater consistency in size and shape and the larger number of alive cells in the training data set.”

4) The model seemed to be validated by visual inspection. However, visual manual approach was criticized earlier in the manuscript. Authors should at least visually verify it with more than one operator. Also, the model could be verified using control 100% dead samples (or other control samples).

5) It is not clear to me how did their ML tool fix aggregates issue at high cell densities, which they mention is a problem. How was the model verified since visually it is hard to count how many cells are in an aggregate?

6) I cannot comment on Model Performance section since I am not an expert in ML field.

7) Data availability: the authors indicated that references 33 and 40 provide the readers the full ML model, access to evaluated images and training data set. However, Ref. 33 website gives an error “DOI not found” and Ref. 40 is not accessible since it asks for “Delft University of Technology GitLab server” login credentials.

Reviewer #2: Finding ways to reduce the cost of scientific research without losing its quality is an important task. This is especially true for countries with weak economies, where scientists often do not have access to expensive equipment, including automatic cell counting systems. Based on the results of their research, the authors of the article proposed a convenient, inexpensive method of automatically counting of cells instead of labour-intensive manual counting. The positive thing is that the proposed method is universal and can be applied to different types of cell cultures. In general, the manuscript submitted for review makes a good impression.

Reviewer #3: The manuscript under review introduces Machine Learning (ML) model aimed at providing an efficient alternative to manual or automated cell counting in Trypan blue-stained microscopic images of insect and HEK cells. The model claims to offer higher speed, lower cost, and the capacity to process numerous images swiftly. It is intended to be beneficial for cell culture operators across all scales and to simplify protein expression requirements in insect cell production platforms.

Despite the promising potential of this work, I have identified several areas of the manuscript that could benefit from improvement:

• The manuscript is well-written but the introduction is excessively long and could be better aligned with the main objectives of the study.

• The provided illustration diagram in the article lacks logical reasoning in directly feeding the sample to the machine learning model. To address this issue effectively, it is crucial to integrate a detection mechanism into the process.

• Reference [3] seems unrelated to the topic at hand. To improve the article's general coherence, it is advised to reevaluate its relevance or eliminate it.

• The ML model that was applied in the study needs to be further explained. It would improve comprehension and provide technical depth to include a detailed discussion of its architecture, important parts, and algorithms.

• There are difficulties with availability and retrieval in reference [15].

• The methodology section (lines 143 to 145) lacks clarity on how the cell counts in the training and validation sets were determined.

• In line 171, the use of the term 'large test data' appears misleading as the number of data points is relatively small for machine learning studies.

• Confusion matrices, which are essential for assessing the performance of ML models, are currently absent from the manuscript and should be included.

• Additionally, I suggest that the authors consider adding the following reference, as it could provide relevant and beneficial insights to their study: DOI: 10.1109/JBHI.2022.3203893

Addressing these points will greatly enhance the overall quality and impact of the manuscript, by providing a more precise representation of the research conducted and its relevance to the field.

6. PLOS authors have the option to publish the peer review history of their article (what does this mean?). If published, this will include your full peer review and any attached files.

Reviewer #1: No

Reviewer #2: No

Reviewer #3: **Yes: **Omer Aydin

---

## [Author Response · Author response to Decision Letter 0]

24 Aug 2023

Our response is contained in the file "Kuijpers et al point-by-point response.pdf".

---

## [Editor Report · Decision Letter 1]

4 Sep 2023

Automated cell counting for Trypan blue-stained cell cultures using machine learning

PONE-D-23-16673R1

Dear Dr. Dekker,

We’re pleased to inform you that your manuscript has been judged scientifically suitable for publication and will be formally accepted for publication once it meets all outstanding technical requirements.

Kind regards,

Abdul Rauf Shakoori

Academic Editor

PLOS ONE
---

## [Editor Report · Acceptance letter]

25 Sep 2023

PONE-D-23-16673R1 

Automated cell counting for Trypan blue-stained cell cultures using machine learning 

Dear Dr. Dekker:

I'm pleased to inform you that your manuscript has been deemed suitable for publication in PLOS ONE. Congratulations! Your manuscript is now with our production department. 

Kind regards, 

on behalf of

Dr. Abdul Rauf Shakoori 

Academic Editor

PLOS ONE